# Human Sperm as an In Vitro Model to Assess the Efficacy of Antioxidant Supplements during Sperm Handling: A Narrative Review

**DOI:** 10.3390/antiox12051098

**Published:** 2023-05-15

**Authors:** Elena Moretti, Cinzia Signorini, Roberta Corsaro, Maria Giamalidi, Giulia Collodel

**Affiliations:** 1Department of Molecular and Developmental Medicine, University of Siena, 53100 Siena, Italy; cinzia.signorini@unisi.it (C.S.); r.corsaro@student.unisi.it (R.C.); giulia.collodel@unisi.it (G.C.); 2Department of Genetics and Biotechnology, Faculty of Biology, University of Athens, 15701 Athens, Greece; m.giamalidi@student.unisi.it

**Keywords:** antioxidants, human sperm in vitro, natural extract, oxidative stress, pathological role of ROS, physiological role of ROS, phytocomplexes, polyphenols

## Abstract

Spermatozoa are highly differentiated cells that produce reactive oxygen species (ROS) due to aerobic metabolism. Below a certain threshold, ROS are important in signal transduction pathways and cellular physiological processes, whereas ROS overproduction damages spermatozoa. Sperm manipulation and preparation protocols during assisted reproductive procedures—for example, cryopreservation—can result in excessive ROS production, exposing these cells to oxidative damage. Thus, antioxidants are a relevant topic in sperm quality. This narrative review focuses on human spermatozoa as an in vitro model to study which antioxidants can be used to supplement media. The review comprises a brief presentation of the human sperm structure, a general overview of the main items of reduction–oxidation homeostasis and the ambivalent relationship between spermatozoa and ROS. The main body of the paper deals with studies in which human sperm have been used as an in vitro model to test antioxidant compounds, including natural extracts. The presence and the synergic effects of different antioxidant molecules could potentially lead to more effective products in vitro and, in the future, in vivo.

## 1. Introduction

Infertility is a global health issue. It is defined as the failure to achieve a pregnancy after 12 months or more of regular, unprotected sexual intercourse [1]. It is estimated that 8–12% of couples worldwide are infertile [2] and resort to fertility medical treatment. Both of the partners of a couple can be responsible for non-conception. In at least 50% of cases, a male infertility factor is involved, isolated or in combination with a female factor [3,4]. There are numerous causes and risk factors contributing to the increasing incidence of male infertility [2]. Many of them share oxidative stress (OS) as a common pathway, including varicocele [5], genitourinary infection and inflammation [6,7]. In this scenario, assisted fertilisation technologies (ART) represent the treatment of choice for many couples facing infertility problems. The use of these techniques implies gamete handling in the laboratory and the exposure of gametes to atmospheric oxygen. In addition, semen laboratory processing such as centrifugation; cryopreservation; exposure to visible light; and the variation in oxygen tension, pH and temperature [8,9] enhance the production of reactive oxygen species (ROS).

Spermatozoa are highly differentiated cells that produce ROS as consequence of aerobic metabolism. Below a certain threshold, ROS play a key role in signal transduction pathways and cellular physiological processes. In particular, spermatozoa ROS are necessary for sperm motility, capacitation, the acrosome reaction and oocyte interaction [10,11]. On the contrary, ROS overproduction in spermatozoa and the consequent antioxidant imbalance can damage the cellular structure [12], including the membrane, particularly rich in polyunsaturated fatty acids (PUFA), and molecules such as DNA and proteins [13,14,15,16].

Assuming that external laboratory conditions are optimal, a strategy to minimise OS during sperm manipulation can include the treatment of the patients with antioxidants even though the real efficacy of in vivo supplementation is still debated [16,17,18]. Another very interesting strategy includes the supplementation of media with antioxidants. Many in vitro studies have investigated the scavenging ability of antioxidant compounds against OS induced in human sperm [9] and many other studies have successfully used antioxidants to supplement media applied in semen cryopreservation [19,20].

Human spermatozoa represent a model for an in vitro study because they are motile, and the motility is a parameter that is easy to evaluate; they are nearly transcriptionally and translationally silenced; they do not have DNA-repair activity; and they lack intracellular antioxidant protection. Thus, they depend on and are deeply influenced by the external environment. This review focuses on human spermatozoa as an in vitro model to test the effect of antioxidants, which could be potentially used in clinical practice during sperm handling, on OS.

## 2. Structure of Human Spermatozoa

Human spermatozoa are highly differentiated and polarised cells made up of a head and a flagellum joined by a connecting piece, all of which is enveloped by the plasma membrane (Figure 1).

The head has an oval and flattened shape, and it comprises the acrosome and the nucleus. The acrosome is a cap-like Golgi-derived vesicle delimited by inner and outer acrosomal membranes and covers about 75% of the head. The acrosome contains lytic enzymes such as hyaluronidase and acrosine, which are necessary for the fertilisation process [21]. Only capacitated sperm can interact with the cumulus oophorous and the zona pellucida of an oocyte, resulting in the acrosome reaction and allowing for sperm to penetrate and fertilise an egg. Sperm can undergo a spontaneous acrosomal reaction before reaching the egg, preventing successful fertilisation. Bowker et al. [22] demonstrated the involvement of a mechanism based on protein acetylation that protects bovine spermatozoa from a spontaneous acrosomal reaction.

The region located at the posterior edge of the acrosome is named the equatorial segment; it plays an important role in the fusion between sperm and egg membranes [23]. The head area underneath the acrosomal cap is called the post-acrosomal region. It is particularly important because it contains the phospholipase Cζ (PLCζ), which is widely considered to be the sperm oocyte activation factor through the stimulation of Ca^2+^ oscillations in the oocyte [24,25].

The core of the sperm head is the nucleus, which contains highly condensed chromatin. During the sperm maturation process, about 85% of histones are replaced by protamines (small proteins rich in arginine) characterised by disulphide bridges between the sulfhydryl groups of cysteines [26,27]. The replacement of histones with protamines allows for dense packaging of chromatin; this organisation plays a pivotal role in protecting the genetic material devoted to the perpetuation of the species.

The head is linked to the tail by the connecting piece, a sort of neck made up of nine striated columns closed by the capitulum, a protein structure that interacts with the head [28]. This protein box encases the proximal centriole, under the nucleus, that shows a typical barrel shape with nine triplets of microtubules and an atypical distal centriole composed of splayed microtubules [29] and originates the axoneme, which forms the core of the entire flagellum (Figure 1). The axonemal structure is regulated by hundreds of microtubule-associated proteins and motor proteins, some of which form helical structures within the microtubule lumen that have been hypothesised to be involved in controlling the direction of sperm motion [30].

The midpiece is the flagellar region surrounded by a mitochondrial helix (Figure 1) that envelopes the axoneme and the nine outer dense fibres (ODFs). Mitochondria play a central role in sperm metabolism and are involved in energy production, redox balance, calcium regulation and apoptotic pathways, all of which are necessary for flagellar motility, capacitation, the acrosome reaction and fertilisation [31]. The midpiece ends with the annulus, a septin-based ring structure located beneath the plasma membrane that connects the midpiece and the principal piece of the mammalian sperm flagellum [32]. The annulus is probably involved in the flagellar assembly during spermiogenesis and confines proteins to define the different regions of the tail in mature sperm.

The principal piece, the longest part of the flagellum, contains the axoneme, ODFs and the fibrous sheath (FS) that wraps the axoneme along the entire length of the segment. The FS is a sperm-specific cytoskeletal structure that acts as a scaffold for enzymes involved in signal transduction and glycolytic pathway [33]. Therefore, the human sperm tail shows a complex anatomy, and every structure (proximal and distal centrioles, mitochondria, the axoneme, ODFs and the FS) plays a significant role in motility. Testicular spermatozoa are immature cells that are unable to fertilize an oocyte. After leaving the testis, spermatozoa transit along the epididymis to acquire motility and fertilizing abilities. During post-testicular maturation, many changes that regard a sperm membrane and sperm proteome profile occur [34]. At this purpose, a peculiar role is played by the epididymosomes, exosomes produced by epididymis that prime spermatozoa with a large amount of proteins and RNAs [35].

## 3. Main Items of OS

One of the fundamental principles of biological processes is the concept of ‘homeostasis’, which is currently defined as a self-regulating process by which a biological system maintains stability while adjusting to changing external conditions. Thus, imbalance among key factors/mechanisms involved in homeostatic regulation can affect health. Oxygen homeostasis is one of the major regulatory mechanisms involved in disease physiology and pathogenesis. In oxygen homeostasis, the oxygen supply needs to be adequate for tissue requirements [36]. The reduced availability of oxygen causes cellular damage, adaptations of biochemical processes with enhancement of the anaerobic pathways and ischaemic or infarct events. Conversely, an increased oxygen concentration can be the cause of pathologies such as the so-called hyperoxic–hypoxic paradox (in ischaemia/reperfusion injury) [37]. The detrimental effects of oxygen are mainly attributable to ROS generation. ROS are oxidants, partially reduced metabolites of molecular oxygen, generated by several different metabolic reactions and cellular processes [38,39]. Such molecular species have also been proposed as second messengers in the activation of several signalling pathways [40,41] and are intimately involved in reduction–oxidation (redox) signalling and macromolecular redox regulation. Oxidants have physiological roles [39] and both oxidant and antioxidant signalling have been proposed as the main features of redox homeostasis [42]. Because of its biological relevance in controlling health conditions and diseases, homeostasis of the redox status has been deeply investigated [42,43]. A fine redox architecture, based on molecular interactions, regulates physiological functions and the balance between oxidants and nucleophiles is preserved in redox homeostasis. In particular, the term oxidative eustress indicates physiological OS involved in maintaining the steady-state set point of redox reactions. Failure to maintain the redox steady-state reference point leads to homeostatic disruption (homeostatic imbalance), which could affect health conditions and lead to diseases of failed homeostasis.

Depending on the level of ROS generation, the efficiency of the antioxidant systems and the interaction of ROS with cellular targets, a condition of oxidative distress may arise as the result of redox imbalance. In particular, OS provokes the establishment of a new radically altered redox steady state. Consequently, oxidative damage can occur because of non-specific reactions between oxidative agents and biological macromolecules (proteins, lipids, nucleic acids and carbohydrates). Initially, the term ‘oxidative stress’ was applied to define a severe pro-oxidant/antioxidant imbalance, in favour of pro-oxidant species, potentially able to cause biological damage [44]. In oxidative eustress conditions, oxidants are present at low levels and react with specific targets for physiological redox signalling. As major redox signalling agents, hydrogen peroxide (H_2_O_2_) and the superoxide anion radical (O_2_^•−^) are continuously produced via physiological cellular metabolism by multiple enzymatic and non-enzymatic processes. H_2_O_2_ signalling primarily occurs through the reversible oxidation of specific thiol groups of protein cysteine residues and results in molecular signalling events involved in phosphorylation cascades, transcriptional regulation, cytoskeletal rearrangements and cell replication [45,46].

Although mitochondria and members of the NADPH oxidase (NOX) family are the best-characterised intracellular sources of ROS, they are also produced by several cellular organelles, including the endoplasmic reticulum and the peroxisomes, as well as by various enzymes, including oxidases and oxygenases, which generate ROS as part of their enzymatic reaction cycles [45]. As general mechanisms of oxidant signalling, NOX enzymes produce extracellular O_2_^•−^, which can spontaneously dismutate to H_2_O_2_ that can diffuse, mainly by aquaporin-dependent pathways, into the cell. Inside the cell, H_2_O_2_ modifies redox activity of specific protein targets by altering cysteine residues. Cysteine residues are thought to be the major redox-dependent switches. Nevertheless, redox reactions can also involve other amino acids, such as methionine, in presence of more powerful oxidants [45].

Interestingly, redox homeostasis appears to greatly reflect the state of electrophile production by lipid peroxidation, in addition to ROS cell flux. During lipid peroxidation, which is a chain of reactions that produce lipid hydroperoxides and their degradation products in membranes, electrophilic activation of the transcription factor nuclear factor erythroid 2-related factor 2 (Nrf2) is the main signalling molecule. H_2_O_2_ is a less efficient activator of Nrf2 compared with thiol-conjugating electrophiles [41]. Nrf2 is a key regulator of the cellular antioxidant response as it controls the expression of genes that counteract oxidative and electrophilic stresses. The relevance of Nrf2 has been discussed in many pathological conditions linked to redox homeostasis imbalance [47]. Additionally, many biomarkers of oxidative damage to proteins, lipids, nucleic acids and carbohydrates have been studied. These evaluations have been joined by the ‘omics’ disciplines that evaluate the genes involved in the modification of the redox homeostasis, paving the way for redox medicine [39].

## 4. Sperm and ROS: An Ambivalent Relationship

Spermatozoa represent a perfect example of the oxygen paradox [48,49]. Indeed, oxygen homeostasis and the maintenance of a redox steady state are critical for spermatozoa. Spermatozoa generate ROS because they need them for several physiological processes (Figure 2). On the other hand, when ROS production overcomes the antioxidant defences, there are detrimental effects to the sperm membrane, proteins and DNA (Figure 2), which have a negative impact on male fertility [50].

### 4.1. ROS and Sperm Physiology

ROS play an important role in sperm physiology because they regulate several intracellular pathways, thus modulating the activation of different transcription factors [50,51,52].

First, ROS are responsible for the stability and compaction of sperm chromatin during epididymal transit and storage. They act as oxidising agents and allow for the formation of disulphide bonds between cysteine residues of protamines, arginine-rich proteins that replace histones during spermiogenesis. Chromatin folding is a crucial event in protecting the paternal genome as the spermatozoa travel through the female reproductive tract [27].

One of the most important roles of ROS in sperm physiology (Figure 2) concerns the process of capacitation by which spermatozoa undergo dramatic changes in the membrane composition and acquire hyperactivated motility, leading to the acrosome reaction and fertilisation [53,54]. The molecular processes behind capacitation include an increase in pH, Ca^2+^ and HCO_3_^−^ influx, efflux of cholesterol from the plasma membrane, an increase in cyclic adenosine monophosphate (cAMP) concentration; and protein hyper-phosphorylation [55,56]. During capacitation, the concentration of O_2_^•−^, H_2_O_2_, nitric oxide and peroxynitrite increase progressively. These ROS stimulate the activation of adenylate cyclase that drives the increase in cAMP. This second messenger activates protein kinase A (PKA) that triggers a massive tyrosine phosphorylation cascade, inhibiting tyrosine phosphatase [57,58]. Redox signalling occurs alongside modification of thiol groups of proteins of the plasma membrane and the inner sperm compartments [53]. As a part of the capacitation process, the hyperactivated motility triggered by calcium signalling [59] is characterised by a high amplitude and extremely asymmetrical beating pattern of sperm flagellum, as well as lateral head displacement. These motility modifications are dependent on ROS-mediated tyrosine phosphorylation of flagellar protein and are needed to make spermatozoa that can penetrate the cumulus oophorous and zona pellucida and fertilise the oocyte [60,61]. In addition, the acrosome reaction, the universal requisite for sperm–egg fusion, is influenced by ROS [62] that increase membrane fluidity. In fact, spermatozoa exposed to H_2_O_2_ show an enhanced acrosome reaction and an increased ability to fuse with oocyte [63]. Finally, the involvement of ROS in sperm function and physiology is supported by the observations that antioxidants can alter sperm maturation and, in particular, catalase or superoxide dismutase can inhibit sperm capacitation and the acrosome reaction [64].

### 4.2. ROS and Sperm Pathology

In recent decades, OS has emerged as a one of the main causes of altered sperm function [65]. Spermatozoa represent an easy target for free radical attack due to the high content of unsaturated fatty acids in their membranes, the limited ability to repair DNA damage and the virtual lack of cytoplasm. Moreover, spermatozoa are poor in antioxidant enzymes such as superoxide dismutase, catalase and glutathione peroxidase, as well as peroxiredoxins [66] that protect most cells from oxidative damage. To compensate the scarce presence of cellular antioxidants, the seminal plasma is rich in antioxidant enzymes and radical scavengers, among them glutathione peroxidase, glutathione-S-transferase, catalase and superoxide dismutase [67]. In addition, other hydrophilic compounds as uric acid, hypotaurine, tyrosine, polyphenols, vitamin C, ergothioneine and glutathione, and hydrophobic scavengers as trans-retinoic acids, trans-retinols, α-tocopherol, carotenoids and coenzyme Q10 are present [68].

The antioxidant properties of seminal plasma are important to balance the presence of ROS that are not just from the physiological production by spermatozoa. Human semen contain various amounts of immature spermatozoa, germinal cells, leucocytes, macrophages and epithelial cells. Among them, the main contributors to OS are leucocytes and immature spermatozoa with large cytoplasmic residues, but in physiological conditions, the redox balance is maintained by antioxidants contained in seminal plasma [69]. However, leucocytospermia occurs when the peroxidase-positive leucocyte concentration exceeds 1 × 10^6^/mL [1]. In this pathology, leucocytes produce a hundred times as much ROS as what they would in physiological conditions and the antioxidant power of the seminal plasma is not sufficient to counteract free radicals, leading to OS [70].

Immature spermatozoa fail to extrude the cytoplasm during maturation and the residual cytoplasm allows for the production of NADPH from glucose-6-phosphate (G6PDH) via the hexose monophosphate shunt [71,72]. NADPH generates ROS by using two different pathways: via NADPH oxidase, a membrane bound enzyme that produces the O_2_^•−^ by oxygen, and via NADPH dehydrogenase, which is responsible for redox reactions in the mitochondria. The enhanced ROS production triggered by immature spermatozoa is responsible for OS propagation to maturing normal spermatozoa during epididymal transit [73].

During recent decades, a growing body of evidence has revealed the role of altered redox balance in seminal plasma, sperm alterations and male infertility [12,72,74]. OS is enhanced in situations when non-physiological ROS levels overwhelm the natural scavenger systems. These situations can be represented by primary pathologies affecting the male reproductive system, including varicocele [75], bacterial and viral infections, inflammation and leucocytospermia; chronic pathologies such as diabetes and cancer [72,76]; and environmental and lifestyle factors such as use of drugs, smoking, pollution and radiation (Figure 2). In these conditions, the unconjugated double bonds of PUFA in the sperm membrane are attacked by ROS, producing lipid hydroperoxides and its secondary decomposition product, aldehydes [71]. These highly reactive by-products produced by lipid peroxidation react with proteins and DNA and alter the proteins of the electron transport chain to induce mitochondrial dysfunction, enhancing the production of mitochondrial ROS in a self-perpetrating mechanism [9,71,77]. The most evident effect of OS and lipid peroxidation on spermatozoa is the loss of motility by inhibiting energy generation and the decrease in vitality as observed when sperm are frozen and thawed, two processes that boost ROS production [71]. The relevance of lipid oxidative damage to sperm conditions has been highlighted by the detection, quantification or immunolocalization of the specific end products of lipid peroxidation (aldehydes and oxygenated metabolites of PUFA) [78,79].

OS is a major cause of DNA damage in mammalian spermatozoa. Aitken and De Iuliis [80] proposed a two-step mechanism for the origin of oxidative DNA damage in spermatozoa. The first phase occurs during spermiogenesis and leads to defective protamination and compaction of sperm chromatin, making DNA more vulnerable to ROS attack. In normal conditions, chromatin compaction is mandatory to protect paternal DNA—this stabilisation makes the spermatozoa resistant to oxidative damage. The second phase is referred to a direct oxidative insult on the DNA due to increased ROS generation by sperm and a loss of extracellular antioxidant protection. The evident ROS effects on sperm nuclear DNA include DNA fragmentation, chromatin cross-linking, base-pair modifications and chromosomal microdeletions [11].

In addition, paternal aging plays a negative effect on sperm parameters and induces a ROS-related DNA fragmentation. These alterations negatively influence the reproductive outcome and offspring health documented for cancer, genetic and congenital diseases, chromosomal alterations and others [81].

## 5. Human Spermatozoa as a Model for In Vitro Studies

There are many reasons why human spermatozoa represent a model for in vitro studies (Figure 3). First, the easy collection of spermatozoa from fertile men can guarantee abundant cellular material. Spermatozoa are differentiated cells with features and specific functions that enable them to reach the oocyte and to fertilise it. The first peculiar characteristic of spermatozoa is their motility, which enables them to reach and fertilise the oocyte. This motility is guaranteed by the ability to obtain energy by both mitochondrial oxidative phosphorylation and glycolysis, the enzymes for which are located along the FS [33].

Spermatozoa were supposed to be incapable of transcription and translation; however, several types of coding and non-coding RNAs have been identified. These RNAs derive from the testis, epididymis, but also by spermatozoa themselves, meaning that also the sperm condensed chromatin can be partially transcribed [82].

However, due to the low level of transcription, sperm are deeply influenced by the external environment [83]. Human spermatozoa do not possess DNA-repair activity. When fertilisation occurs, DNA-repair activity depends on the oocyte transcripts that had been stored during maturation [84]. Finally, due to an intrinsic lack of intracellular antioxidant protection, human spermatozoa have a limited capacity to repair oxidative damage [12].

These aforementioned characteristics make spermatozoa a general and ideal cell model to test in vitro many compounds at different concentrations related and unrelated to the reproductive field (Figure 3). Indeed, many antioxidants [85,86,87,88,89,90,91] have been tested using this model. In addition, spermatozoa have been used as an in vitro monitor of toxicity due to many compounds including natural substances that have potential contraceptive activity; thus, this research has aimed to identify promising products that could be used as vaginal contraceptive agents [92,93,94,95].

There is a broad group of studies dealing with human sperm as model for testing the potential toxic effects of pollutants as heavy metals and phthalates [96,97,98], nanoparticles [99,100,101], herbicides [102] and drugs [103,104,105]. When these protocols to study the effects of spermatozoa exposure to compounds of interest are applied, it is important to evaluate how the different sperm structures react to the treatment (Figure 3).

The functional status of the acrosome can be tested by assessing several molecular biomarkers such as acrosin, equatorin, A disintegrin and metalloprotease 3 (ADAM3) and others [106]. However, fluorescently labelled lectins, such as *Arachis hypogaea* agglutinin (PNA, peanut agglutinin), *Pisum sativum* (PSA) and *Canavalia ensiformis* (Con A), represent an easy and inexpensive method to visualise the morphology of the acrosome and the acrosome reaction to determine the percentage of spermatozoa undergoing exocytosis upon stimulation [107]. Lectins interact with specific carbohydrates and provide information on the morphology of the acrosome rather than on the molecules that function during the fertilisation process. The analysis can be performed with flow cytometry or by scoring cells using a light microscope equipped with a fluorescence apparatus.

Another endpoint that can be evaluated after in vitro treatment of spermatozoa is DNA integrity. The World Health Organization (WHO) guidelines [1] report the most common tests to assess sperm DNA integrity. The direct techniques are terminal deoxynucleotidyl transferase deoxyuridine triphosphate (dUTP) nick-end labelling (TUNEL) and a single-cell gel electrophoresis assay (comet assay). TUNEL enables detecting DNA fragmentation by labelling the 3′-OH generated by the breaks with fluorescent nucleotides. The comet assay is a gel electrophoresis-based method that can be used to measure DNA damage in individual spermatozoa. An indirect test based on acridine orange measures the susceptibility of the DNA to denaturation under acidic conditions. The evaluation can be performed by flow cytometry (sperm chromatin structure assay [SCSA]) or by fluorescence microscopy [108].

Mitochondria, the hallmark of the sperm tail midpiece, control motility, ROS production, redox equilibrium and calcium regulation, represent an important indicator of the health status of spermatozoa. An easy and fast way to measure the mitochondrial membrane potential is the use of the fluorescent cationic dye 5,5,6,6′-tetrachloro-1,1′,3,3′-tetraethylbenzimi-dazoylcarbocyanine iodide (JC-1) that shows membrane potential-dependent accumulation in the mitochondria [109]. JC-1 forms J-aggregates and fluoresces red when the mitochondrial membrane potential is high, while it remains in its monomeric state, emitting green fluorescence, when the membrane mitochondrial potential is low.

Finally, sperm motility is one of the most important indices of cell function. A reduction in motility indicates the presence of damage caused by the compounds used in in vitro studies. Analysis of sperm motility is easy and cheap: it requires a light microscope and a cell-counting chamber [1]. In addition, a sophisticated method such as computer-assisted sperm analysis (CASA) systems represents a great tool that has the ability to provide the rapid, reliable and objective quantitative assessment of sperm kinematic characteristics [110].

Vitality should be assessed concomitantly with motility, particularly when the in vitro treatment reduces this parameter drastically. The evaluation of both motility and vitality allows for clarifying whether the immotile spermatozoa are dead or alive, and this is important to understand whether the substance under study has an impact on sperm motility or if it is able to kill the spermatozoa. Eosin-nigrosin stain and the hypoosmotic swelling test are the most commonly applied assays. Eosin-nigrosin staining is based on the integrity of the plasma membrane: dead spermatozoa are stained because the membrane is disrupted. The hypoosmotic swelling test is based on the semi-permeable properties of the plasma membrane: living spermatozoa with intact membranes swell in hypotonic solutions [111].

## 6. Human Sperm as a Model to Assess the Antioxidant Activity of Different Compounds

The literature is quite rich with papers exploring the use of human sperm as in vitro cellular model in testing antioxidant compounds. This field of research was born from the need to protect spermatozoa from OS during semen handling and ART. The idea was to engineer and supplement incubation media with antioxidants in order to preserve the functional integrity of spermatozoa.

In humans and animals, sperm cryopreservation represents an important strategy to store semen for different purposes. Human semen is conserved for assisted fertilisation—for example, if a man must receive chemotherapy or for sperm banks. In animals, semen cryopreservation is important in order to use reproductive biotechnology (artificial insemination and in vitro fertilisation), which is extensively applied to preserve genetic resources, enhance male fertility and control diseases [112]. In both cases, even though the cryopreservation protocols can supply many benefits, they impair spermatozoa function [113]. Within this wide topic, we restrict our focus to human sperm treated in vitro with antioxidants.

Pentoxifylline (Table 1) was one of the first substances considered in this kind of studies [114,115,116,117,118,119,120,121,122], particularly from 1990 to 2000. Pentoxifylline and the other compounds reported in Table 1 showed a general positive effect on sperm motility, antioxidant activity and in protecting human sperm from elevated OS generated during cryopreservation protocols. Pentoxifylline is a methylxanthine derivative that may influence human sperm respiration, motility and acrosome reaction by increasing the intracellular cAMP concentration and nitric oxide level that stimulate the guanylate cyclase (cGMP) pathway [123,124].

In addition, the antioxidant properties of pentoxifylline are due to the inhibition of xanthine oxidase, of which causes a reduction in both the intracellular ROS level and lipid peroxidation [117]. The effects of pentoxifylline on ART outcomes are available, but the findings remain unclear [125,126,127,128].

Some authors reported that an indiscriminate use of pentoxifylline does not improve in vitro fertilization performance due to a premature acrosomal reaction [122,125,126].

Quite surprisingly, pentoxifylline was used in the case of systematic sperm defects in which motility was severely compromised by genetic mutations. It was described as a mild positive effect of this compound on sperm motility in case of Kartagener syndrome [129,130]; however, we believe that the use of this compound in the presence of systematic defects is not exactly appropriate.

Incubation with papaverine [131], zinc, aspartate and coenzyme Q [132], selenium [133] and L-carnitine [134] protected sperm motility, whereas the effect of vitamin E [135,136,137] and ascorbic acid [137,138] on sperm motility is debated (Table 1). Among these compounds, zinc is able to inhibit in vitro superoxide anion generation [132,139,140]; Ajina et al. [141] reported in vitro an antioxidant effect on the sperm parameters of infertile men, but no effect on sperm lipid peroxidation. Coenzyme Q10 (CoQ10) is a lipophilic molecule present in mitochondrial respiratory chain able to transfer electrons from complex I and II to complex III. Exogenous CoQ10 may diffuse into the polyunsaturated lipid bilayer of the plasma membrane, enhancing energy production and preventing ROS generation [142]. Boonsimma et al. [143] supplemented with CoQ10 semen samples of asthenozoospermic men and observed a positive effect of on sperm motility, but OS was not reduced.

Among the considered compounds, vitamin E [144,145,146,147] and its analogues [136] as well as ascorbic acid have a relevant role in counterbalancing the negative effect of ROS and membrane lipoperoxidation [138,148].

As mentioned above, lipid peroxidation is a process in which lipids are oxidized as consequence of failed redox steady-state regulation. Membrane lipid peroxidation causes a rapid progressing oxidative degradation of lipids by a chain reaction that proceeds in three stages: initiation, propagation, and termination [149]. In the first stage, the first lipid radical, initiating the peroxidative chain reaction, is formed; along the second stage, further lipid radicals are formed by a chain reaction involving more and more lipid molecules; finally, termination of the peroxidative chain reaction occurs by radical–radical interaction or by the intervention of antioxidants, such as vitamin E (α-tocopherol). Vitamin E is a lipid-soluble molecule able of blocking lipid oxidative chain, donating its hydrogen to a lipid radical (making it inactive) and forming the vitamin E radical. Thus, vitamin E increases and maintains membrane fluidity by protecting oxidizable lipids. Among the family of vitamin E isoforms, α-tocopherol is the most largely investigated compound in different human conditions, in that α-tocopherol appears to reverse human deficiency symptoms and to be preferentially retained by the body [150,151]. Furthermore, vitamin E has been reported to be involved with many enzyme activities, also including phospholipase A_2_ and lipoxygenases, which are involved in the biosynthesis of lipid mediators. Moreover, vitamin E is involved in cellular signalling and is able to modulate the activity of protein kinase C, a key regulator in signalling pathways.

Antioxidant properties have also been demonstrated for S-glutathione and hypotaurine [152], oleoylethanolamide [153], ethylenediaminetetraacetic acid and catalase [154], curcumin [87] and lycopene [155,156] (Table 1).

Recently, the most commonly used substances in in vitro studies and applications on freezing protocols have been inositol/myoinositol [157,158,159,160,161,162,163,164,165] and, in particular, melatonin [166,167,168,169,170,171,172]. The effects of these compounds are reported in Table 1.

Inositol is a component of the vitamin B complex. Myoinositol is the most biologically important form in nature and is involved in the mechanisms of signal transduction in the plasma membrane as precursor of second messengers. Myoinositol increases cytosolic Ca^2+^ and, consequently, increases mitochondrial Ca^2+^ that stimulates the oxidative mechanism and the ATP production, improving the mitochondrial function of spermatozoa, preventing apoptosis and facilitating chromatin compactness [158].

After sperm thawing, it has been reported that myoinositol induces a significant increase in oxygen consumption and a significant decrease in the level of carbonyl groups, the main structural changes occurring in the conditions of OS [164].

Human seminal fluid contains melatonin, and melatonin receptors are expressed on the sperm plasma membrane [173]. The vitality of human spermatozoa is significantly improved via exposure to 1 mM melatonin for 30 min [174]. In addition, melatonin can reduce OS and increase the expression of heat shock protein 90 (HSP90) in spermatozoa that are subjected to cryopreservation [175]. Two research groups observed that melatonin exerted anti-apoptotic activity in spermatozoa by reducing caspase-3 activation [176] and DNA fragmentation [171].

Pentoxifylline [123,177,178,179], zinc [180,181], tocopherols [182,183,184,185,186], ascorbic acid [187,188,189], *N*-acetylcysteine [190,191], L-carnitine [190,192,193,194,195], glutathione [196,197], hypotaurine [198], catalase [188,199,200], coenzyme Q [194] and curcumin [201,202,203] have been used in freezing protocols. Overall, they have had generally positive activity on human sperm quality (Table 1).

**Table 1 antioxidants-12-01098-t001:** In vitro effects of compounds with antioxidant activity on human sperm. The table shows the effects (negative effects are in bold) of the in vitro treatment of human spermatozoa with different antioxidant compounds. Clinical studies were not considered. Regarding the column related to freezing protocols, letter “b” means that the supplementation was given before freezing, letter “a” means that the supplementation was given after thawing.

Investigated Compounds	Effects	Freezing Protocol
Pentoxifylline	Motility positive effect [114,116,118,119,120,121,122] * Antioxidant activity [115,117]	[123] b, [177] b, [178] ab, [179] b
Papaverine	Motility positive effect [131]	[131] a
Zinc	Motility positive effect [132] Antioxidant activity [132,139,140,141]	[180] b, [181] b
Selenium	Motility positive effect [133] Antioxidant activity [133]	
Vitamin E (α tocopherol, liposoluble) Trolox (hydrosoluble)	Motility positive effect [135,136] **Motility negative effect** [137] Antioxidant activity [135,144,145,146,147]	[182] b, [183] b, [184] b, [185] b, [186] b
D-aspartate	Motility positive effect [132] Antioxidant activity [132]	
Ascorbic acid	Motility positive effect [138] **Motility negative effect** [137]; Antioxidant activity [138,144,148]	[187] b, [188] b, [189] b
*N*-acetylcysteine		[190] b, [191] b
S-glutathione	Antioxidant activity [152]	[196] ab, [197] b
Hypotaurine	Antioxidant activity [152]	[198] b
Oleoylethanolamide	Motility positive effect [153] Antioxidant activity [153]	
Ethylenediaminetetraacetic acid (EDTA)	Motility positive effect [154] Antioxidant activity [154]	
Catalase	Motility positive effect [154] Antioxidant activity [154]	[188] b, [199] b, [200] b
Coenzyme Q	Motility positive effect [132,143] Antioxidant activity [132,144]	[194] b
L-carnitine	Motility positive effect [134]	[190] b, [192] b, [193] b, [194] b, [195] b
Phosphatidylcholine		[195] b
Myoinositol/Inositol	Motility positive effect [157,158,159,163]	[160] b, [161] b, [162] b, [164] ab, [165] b
Melatonin	Motility positive effect [166,170,171,172,174,176] Antioxidant activity [166,171,172] Anti-apoptotic effect [171,176]	[167] b, [168] b, [169] ab, [175] b
Curcumin	Antioxidant activity [87]	[201] b, [202] b, [203] b
Lycopene	Antioxidant activity [155,156]	

* Pentoxifylline analogue.

In the last two decades, there has been growing interest in using polyphenols for cryopreservation. Polyphenols, natural compounds synthesised exclusively by plants, are an integral part of the human diet; they are present in fruits, vegetables, nuts, seeds, tree barks, and beverages such as wine, beer and tea. Polyphenols include the subclasses of non-flavonoid polyphenols—for example, resveratrol—and flavonoid polyphenols such as quercetin, epigallocatechin-3-gallate, genistein, naringenin and many others [204].

It is known that flavonoids can play a dual action on ROS homeostasis as they can behave as antioxidants under normal conditions, but they can also have pro-oxidant activity; this is the reason why they can trigger apoptotic processes in cancer cells [205]. Several mechanisms have been suggested to explain the biological activity of polyphenols on animal cells. Polyphenols directly scavenge ROS and chelate metal ions due to the presence of hydroxyl groups, and they can show specific actions strictly dependent on the particular structural and chemical characteristics of the different molecules. These indirect effects include the activation of antioxidant enzymes, suppression of pro-oxidant enzymes and others [206].

Polyphenolic compounds have been used in in vitro studies with human sperm and in cryopreservation protocols (Table 2). Due to the powerful antioxidant activity [207,208], these substances show a general protective effect on sperm motility [85,209,210], vitality [88,211,212], membrane and acrosome integrity [213,214] and the DNA status [145,208,215]. To the best of our knowledge, quercetin, genistein and resveratrol have also shown protective effects on OS triggered by the cryopreservation protocol [187,216,217,218,219,220,221,222,223].

However, Aitken et al. [224] demonstrated that many polyphenols could be harmful for human spermatozoa and suppress motility by different mechanisms of action. In this research, human spermatozoa were incubated for 24 h with several polyphenols (a concentration range of 25–200 µM), including quercetin, catechin, gossypol, gallocatechin gallate, catechin gallate, caffeic acid, pyrogallol, epigallocatechin gallate, resveratrol, epicatechin gallate, genistein, baicalein and ellagic acid. In terms of toxicity, 50 µM of gossypol caused a rapid loss of motility due to different mechanisms of action. Gossypol can bind the proteins of the mitochondrial electron transport chain, causing a leakage of electrons that react with oxygen producing superoxide anion, and, in particular, gossypol can covalently modify axonemal proteins needed for motility. In addition, a rapid loss of the mitochondrial membrane potential and a dramatic increase in ROS production were observed [224]. For most compounds examined in this study [224], the loss of sperm motility was gradual and correlated with an increase in lipid peroxidation, indicating the pro-oxidant activity of polyphenols. Other polyphenols as genistein showed completely different behaviour, inducing high levels of mitochondrial redox activity that was not concomitant with increased lipid peroxidation. Genistein can interact with complex III (instead of complex I) of the sperm mitochondrial transport chain, causing electron leakage; this process did not damage spermatozoa. Of note, resveratrol and genistein at a concentration of <100 µM appeared to be more tolerated by human spermatozoa (Table 2). There have also been conflicting results regarding the ability of polyphenols to protect sperm from OS in other studies. In one study, quercetin, rutin, naringenin and epicatechin exerted a dose-dependent effect (20–400 µM) on sperm motility and vitality [213]. At a high concentration, quercetin was toxic even though at 30 µM it was well tolerated by spermatozoa and exhibited antioxidant activity. Khanduja et al. [225] also observed that quercetin exerted a dose-dependent effect on sperm motility. Naringenin was not toxic for sperm up to a concentration of 200 µM; however, at 400 µM, it caused sperm death. Epicatechin at 400 µM caused a decrease in progressive sperm motility and an increase in non-progressive sperm motility [213]. Recently, Lv et al. [226] exposed in vitro human sperm to different concentrations of rosmarinic acid (1, 10, 100 and 1000 μM) and observed a dose-dependent decrease in sperm motility, capacitation and the spontaneous acrosome reaction.

**Table 2 antioxidants-12-01098-t002:** Effects of flavonoids and polyphenols used in in vitro experiments on human spermatozoa. The table shows the effects (negative effects are in bold) of the in vitro treatment of human spermatozoa with different flavonoids and polyphenols. Clinical studies were not considered. Regarding the column related to freezing protocols, letter “b” means that the supplementation was given before freezing, letter “a” means that the supplementation was given after thawing.

Investigated Flavonoids and Polyphenols	Effects	Freezing Protocol
Rosmarinic acid	**Negative effect on motility and acrosome reaction** [226]	
Quercetin	**Negative effect on motility** [225] Positive effect on sperm motility [210,213,214] * Positive effects on membrane integrity, sperm vitality, acrosome [213,214] * Positive effect on DNA integrity [145] **Negative effect on DNA integrity** [214] * Antioxidant activity [85]	[219] b, [220] b, [223] b
Genistein	Positive effects on membrane integrity [224]	[217] a
Equol	Positive effect on DNA integrity [215]	
Rutin	Positive effect on sperm motility, membrane integrity, sperm vitality [213] Positive effect on DNA integrity [145]	
Naringenin	Positive effect on sperm motility, membrane integrity, sperm vitality [213] Positive effect on DNA integrity [145]	
Epigallocatechin-3-gallate	Antioxidant activity [207]	
Hydroxytyrosol	Positive effect on sperm motility, vitality, DNA integrity [211] Antioxidant activity [211]	
Caffeic acid phenethyl ester	Positive effect on DNA integrity [208] Antioxidant activity [208]	
Procyanidine	Positive effect on sperm motility [209]	[209] b
Ellagic acid	Positive effect on sperm motility, vitality, DNA integrity [212]	
Resveratrol	Positive effect on sperm motility [85,218] Positive effect on membrane integrity [224] Antioxidant activity [85]	[187] b, [216] b, [218] b, [221] b, [222] b
Cholorogenic acid	Positive effect on sperm motility, vitality, DNA integrity [88]	[88] b

* quercetin-loaded liposomes.

We cannot exclude that the modulation in positive and negative ways of the sperm motility by flavonoids can involve oestrogen (ER) receptors that are present in human spermatozoa; in particular, ERα is predominantly expressed in the midpiece and ERβ is distributed along the tail [227]. Indeed, several flavonoids interact with ERs, although with low affinity, and they behave both as oestrogen antagonists or oestrogen agonists; however, the mechanisms underlying the contrasting effects are still poorly understood [228].

In addition, there was DNA damage when quercetin-loaded liposomes were incubated in vitro with human spermatozoa, indicating possible toxic activity of this compound [214]. Quercetin seems to have weak mutagenic activity in in vitro experiments and its toxic effect against cancer cells is well documented [229], depending on the dose at which it is used [230].

The debated effects of polyphenols on human sperm in vitro could limit their clinical application. Hence, there is a need for studies focused on the potential genotoxicity of these compounds and on the exact mechanism of action before they can be used in gamete handling procedures.

In the last decade, attention has shifted to natural extracts, including phytocomplexes (Table 3). The rationale behind using a phytocomplex rather than its isolated constituents is based on the synergic effects between the molecules and increased bioavailability in the in vivo studies [231].

Table 3 reports the studies in which an aqueous [232,233,234,235,236,237,238] or alcoholic [91,239,240,241,242] extract of plants or natural products was applied in vitro and during cryopreservation protocols to test the protective/antioxidant effects on human spermatozoa. Because the botanicals and herbal preparations for experimental or medical use contain different types of bioactive compounds, chemical characterisation of the extract and evaluation of the composition, antioxidant activity and toxicity are mandatory [91,218,235,236,241,242,243,244,245].

Other compounds such as *Cissampelos capensis* rhizome aqueous extract [246] and unfermented rooibos [247] seem to induce sperm capacitation in vitro. Some researchers have analysed the effects of phytocomplexes on sperm DNA [238,240,244,245,248]. The authors have reported a general protective action on DNA.

Although the limited data on the use of these extracts during cryopreservation procedures [218,249,250,251,252] showed positive results on sperm motility, vitality and DNA fragmentation, it is impossible to compare the results even if they are related to the same botanical species because of the different origin of the plants and extraction methods.

**Table 3 antioxidants-12-01098-t003:** Natural extracts that exert a positive effect on human sperm in vitro. The table shows the papers that report the positive effects of the in vitro treatment of human spermatozoa with different natural extracts. Regarding the column related to freezing protocols, letter “b” means that the supplementation was given before freezing, letter “a” means that the supplementation was given after thawing.

Natural Extract	Extract Characterization and Effects on Human Sperm	Freezing Protocol
Propolfenol^®^	Extract characterization, antioxidant activity [239]	
Chilean propolis ethanolic extract	Positive effect on DNA integrity, antioxidant activity [240]	
*Withania somnifera* aqueous ethanol extract	Extract characterization, positive effect on sperm parameters, antioxidant activity [91]	
*Prunus japonica* seed ethanolic extract	Extract characterization, positive effect on sperm parameters [242]	
*Moringa oleifera* aqueous extract	Positive effect on sperm parameters, DNA integrity, antioxidant activity [238]	
*Capparis spinosa* L. hydroalcoholic extract	Extract characterization, positive effect on sperm parameters, DNA integrity [245]	
*Origanum vulgare*	Positive effect on sperm parameter [237] essential oil obtained by hydrodistillation	[251] b aqueous extract
*Castanea sativa* Mill. ethanolic extract	Extract characterization, positive effect on sperm parameters, antioxidant activity [241]	
*Eruca sativa* aqueous extract	Extract characterization, positive effect on sperm parameters, antioxidant activity [236]	
*Tribulus terrestris* aqueous extract	Positive effect on sperm parameters [234]	[249] ab
*Terminalia arjuna* bark aqueous extract	Extract characterization, positive effect on sperm parameters, antioxidant activity [235]	[250] b
*Morinda officinalis*	Extract characterization, positive effect on DNA integrity, antioxidant activity [244]	
Date seed oil	Positive effect on sperm parameters, DNA integrity, antioxidant activity [248]	
*Mondia whitei* aqueous extract	Positive effect on sperm parameters [233]	
Aqueous extract of herbal medicines	Positive effect on sperm parameters [232] *	
The red alga *Gelidiella acerosa*	Extract characterization, positive effect on sperm parameters [243]	
*Opuntia ficus-indica*	Extract characterization [218]	[218] b, [252] a

* Astragalus membranaceus and Acanthopanacis senticosi.

## 7. Conclusions and Future Directions

OS can have a negative impact on gametes during the laboratory procedures that occur in the presence of oxygen [9,12,16]. ROS generation during these procedures cannot be completely avoided; hence, strategies minimising oxidative damage are advisable. One of the most important approaches involves using antioxidants to supplement the media used for gamete handling and to optimise gamete and embryo preservation [16,88,128].

Based on their intrinsic characteristics and the easy ability to collect, spermatozoa represent a valuable in vitro cellular model to test the effect of antioxidants against OS. The standardisation of this type of study would be important. When new molecules are tested, the main endpoints of spermatozoa as motility, vitality, the status of the acrosome, the membrane mitochondrial potential and, in particular, the DNA status and the potential cytotoxicity should be considered and evaluated.

The finding of the best antioxidants and their optimal concentrations to be used during the in vitro supplementation of media used for semen handling is an open area of research. Regarding the use of antioxidants during freezing–thawing protocols, to date, the supplementation during freezing appeared to be the most used method; however, some research reported a supplementation after thawing or both during cryopreservation and post-thawing (Table 1, Table 2 and Table 3).

Most of the studies in this field are essentially observational and new insights into the mechanism of action of the various compounds are necessary. Many compounds with antioxidant activity have been tested in vitro on human spermatozoa [147,154,159,171,213,224]; however, in the last decade, there has been increasing interest in phytochemicals and natural extracts [91,218,232,233,234,235,236,237,238,239,240,241,242,243,244,245,246,247,248,249,250,251,252]. These extracts are obtained from waste materials that represent a rich source of molecules with high antioxidant activity. This particular field of research is worth implementing because the by-products can be utilised in the industrial, cosmetic, nutraceutical fields and other areas, encouraging recycling and moving from a linear economy to a circular, sustainable green economy [253].

## Figures and Tables

**Figure 1 antioxidants-12-01098-f001:**
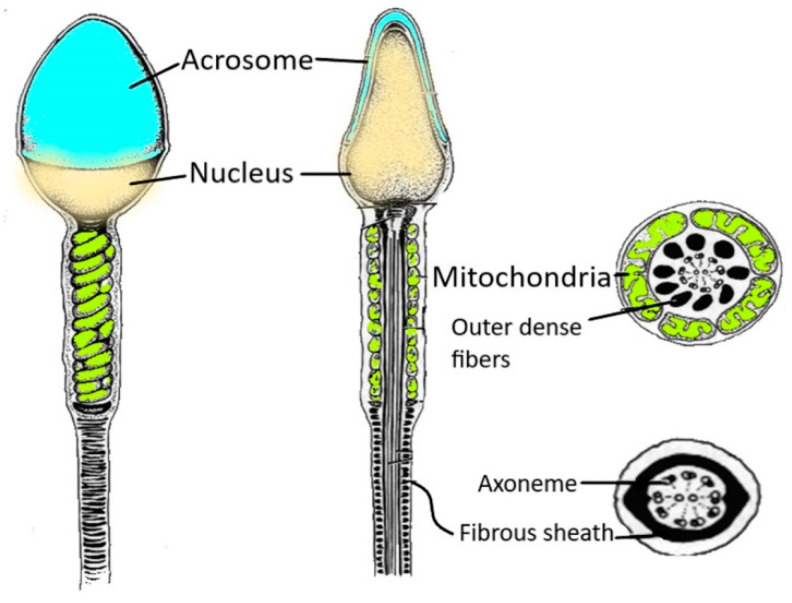
Spermatozoon structure. The figure shows the various regions of a human spermatozoon. Starting from the head region, it is possible to distinguish the acrosome and the nucleus. The sperm flagellum contains the axoneme and periaxonemal structures such as outer dense fibres and a fibrous sheath.

**Figure 2 antioxidants-12-01098-f002:**
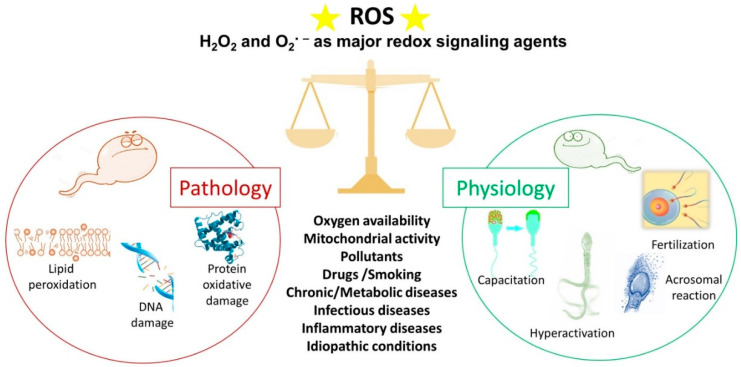
The double face of reactive oxygen species (ROS). Under physiological conditions, ROS regulate the sperm mechanisms involved in capacitation, hyperactivation, the acrosome reaction and fertilisation. On the contrary, some pathophysiological conditions can increase ROS levels over the physiological threshold, generating a condition of oxidative stress that leads to lipid peroxidation, DNA damage and protein oxidative damage. The stars represent ROS.

**Figure 3 antioxidants-12-01098-f003:**
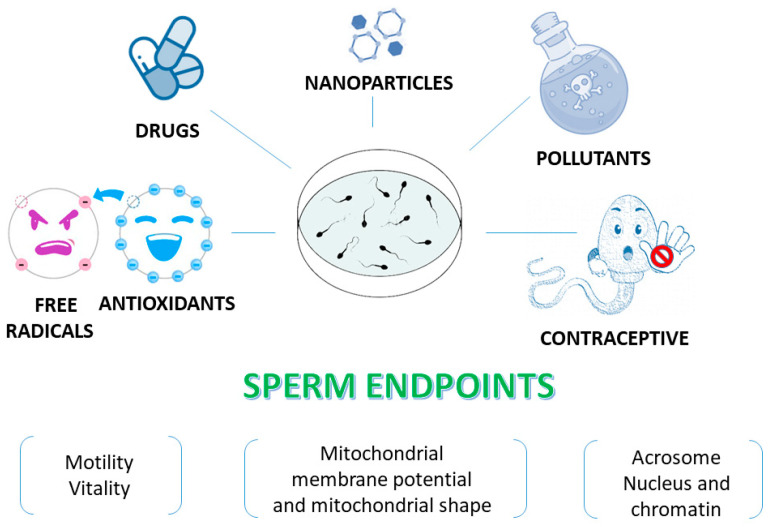
Human sperm as an in vitro model to test various compounds. Sperm endpoints such as motility, vitality, mitochondrial membrane potential, the acrosome status and DNA integrity can be evaluated when human spermatozoa are used as an in vitro model to test effects of toxic compounds, natural extracts and molecules.

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
