# Peer review of "Human Sperm as an In Vitro Model to Assess the Efficacy of Antioxidant Supplements during Sperm Handling: A Narrative Review"

_antioxidants, 2023, doi:10.3390/antiox12051098_

Round 1

Reviewer 1 Report

The manuscript constitutes the elaboration on spermatozoon biology and the potential use of this cell as a model cell. In my opinion, the manuscript can be enriched with some more information.

I am not sure the title reflects the content of the ms

“human spermatozoa represent a valid model …..”- provide references. Also there are some reports showing that spermatozoa can conduct .transcription, please discuss it

“transcriptional and translational activities are virtually silenced in spermatozoa; the function of these cells is essentially regulated at the protein level”- please give information about proteins expressed by sperm including sex steroid hormone receptors and the origin of these proteins (e.g. epididymis, proteosomes etc).

Information about Kartagener syndrome should be added to section 2

There should be also a description of how spermatozoon mature in the epididymis, what functional parameters will occur in spermatozoa after that

What about spermatozoa aging and diseases carried from the father's side?

Minor editing of English language required

Reviewer 2 Report

Dear the Editor

Moretti E et al reviewed a series of studies involved in the efficacy of in vitro antioxidant testing using human sperm and the discussion for future expansion of in vivo application such as cryopreservation. This is a well-written manuscript covering most items which need to be included under this subject. 

Major issue:

One thing that might be added would be a short discussion for lipid peroxidation and the role of a-tocopherol. As aware, these were classical, but extensively studied example of ROS reaction, particularly when these authors would like to refer the role of flavonoids (Table 2) as well as some antioxidants (Table 1, vitamin E/trolox, ascorbic acid, coenzyme Q).

Minor concerns:

1) Is there no ethical problem to perform these assays using human sperm?

2) In page 6, attacked by ROS, producing aldehyde” should be  "... producing lipid hydroperoxides and its secondary decomposition product aldehydes"

Typos:

Page 2, section 3, last para, "have been studies studied" reads "have been studied".

Reviewer 3 Report

The authors review studies using human spermatozoa as an in vitro model to test the effects of antioxidants in sperm handling. This topic is interesting and the manuscript is relatively easy to follow. However, the discussion is very superficial and does not provide a critical assessment of the literature. There are two major issues that need to be addressed before the manuscript can be considered for publication.

1. Section 8 resembles a catalog. The authors listed a number of compounds with possible antioxidant activity and described their effects on human spermatozoa. However, for many compounds, they do not go deeper to discuss the mechanisms underlying their protective activity. In particular, for compounds that show both positive and negative effects, the authors need to provide some explanations. Thus, this section should be rewritten to provide a mechanistic analysis of compounds with antioxidant activity on spermatozoa.

2. The tables are not well presented. It would be more clear-cut if the authors indicate simply positive or negative effect followed by citing the literature according the journal style (I do not think it is necessary to list the authors). Compounds that show negative or no effect were indicated in bold, but for the readers, it is unclear whether they have negative effect or have no effect. I recommend the authors to reorganize the tables and remove those studies showing no effect.

The quatilty of English language is good.

Round 2

Reviewer 1 Report

My suggestions were included in the corrected text

Author Response

Thank you

Reviewer 2 Report

Dear the Editor

Raised concerns by the Reviewer have been properly addressed.

Author Response

Thank you

Reviewer 3 Report

The manuscript is significantly improved, but for different tables I would like a brief description of the methods used for freezing protocol, for example, whether compounds/extracts were added before cryopreservation or supplemented after the freeze-thaw process (see reference 249). For table 3, the columns “extract characterization” and “effects” may be combined since they cite the same references. Otherwise, there is a need to provide a description about how these natural extracts were characterized.

Since the tables are modified, the sentence “Last column shows the researches in which the different natural extracts were used as supplement during freezing protocols” is no longer appropriate.
